# DexCatch: Learning to Catch Arbitrary Objects with Dexterous Hands

**Fengbo Lan**[1,*]    **Shengjie Wang**[1,2,3,*]    **Yunzhe Zhang**[1]
**Haotian Xu**[1]    **Oluwatosin Oseni**[4]    **Ziye Zhang**[1]    **Yang Gao**[1,2,3,†]    **Tao Zhang**[1,†]

[1]Tsinghua University    [2]Shanghai Artificial Intelligence Laboratory
[3]Shanghai Qi Zhi Institute    [4] Colorado School of Mines
∗ Equal contribution    † Corresponding author

**Abstract:** Achieving human-like dexterous manipulation remains a crucial area of research in robotics. Current research focuses on improving the success rate of pick-and-place tasks. Compared with pick-and-place, throwing-catching behavior has the potential to increase the speed of transporting objects to their destination. However, dynamic dexterous manipulation poses a major challenge for stable control due to a large number of dynamic contacts. In this paper, we propose a Learning-based framework for Throwing-Catching tasks using dexterous hands (LTC). Our method, LTC, achieves a 73% success rate across 45 scenarios (diverse hand poses and objects), and the learned policies demonstrate strong zero-shot transfer performance on unseen objects. Additionally, in tasks where the object in hand faces sideways, an extremely unstable scenario due to the lack of support from the palm, all baselines fail, while our method still achieves a success rate of over 60%. Video demonstrations of learned behaviors and the code can be found on the supplementary website.

**Keywords:** Reinforcement Learning, Dexterous Manipulation, System Stability

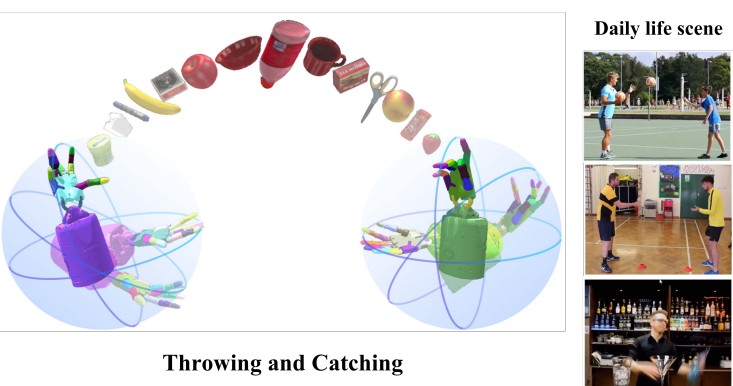

Figure 1: Generalization tasks for throwing and catching objects with shadow hands. Shadow hands can throw and catch a variety of objects with different shapes, masses, and other properties when the hands face sideward.

## 1 Introduction

The study of dexterous hands is a widely pursued research area aimed at matching the intricate capabilities of human hands. Researchers employ these hands for delicate manipulations [1, 2, 3, 4]. Recently, researchers have applied reinforcement learning (RL) techniques to various tasks involving dexterous hands, leading to diverse outcomes [5, 6, 7]. They have tackled a series of manipulation

8th Conference on Robot Learning (CoRL 2024), Munich, Germany.

tasks, including in-hand reorientation [6, 8], opening a door [5], assembling LEGO blocks [9], and even solving a Rubik's cube [7]. These tasks can be summarized as common behaviors in daily living, where hands pick up an object, reorient it in hand, and then either place it or use it as a tool [8].

While pick-and-place can fulfill the requirements of many daily tasks, dynamic throwing-catching holds the potential for more efficient task execution [10, 11, 12]. For example, two robots can employ throwing and catching to efficiently transfer objects instead of pick-and-place, as shown in Fig.1. Meanwhile, it can extend the working space of robots, which avoids the potential collisions for robot interactions [10]. However, precisely catching arbitrary objects proves to be highly challenging [13]. 1) First, the dynamic contact with strong force makes the object fall from the hand. In previous work [13], model-free reinforcement learning methods achieved nearly 0% success rate in the throwing-catching task within 30 million environment steps. 2) Secondly, varying object-centric properties (e.g. mass distribution, friction, shape) require the catching behavior robust enough, especially for objects of daily life. To target this problem, previous methods must learn the dynamics of a moving object [14]. The additional module requires expert demonstrations and increases the system's complexity. 3) Thirdly, when faced with flying objects coming from random directions, the catching hand must adopt different postures, enabling a larger working space. However, the challenge arises in catching objects with hands facing sideways, where objects easily fall due to the lack of any supporting surface, as illustrated in Fig. 1. Due to this challenge, the task of catching objects with hands facing sideways has not been explored previously.

To address the aforementioned challenges, we propose a **L**earning-based framework for **T**hrowing-**C**atching tasks using dexterous hands (LTC). The simple but effective framework can achieve superior performance across diverse daily objects, in various throwing and catching scenarios where the hands face upward or sideward. The main contributions of this work are as follows.

- We propose a simple and effective framework using a model-free reinforcement learning algorithm, to solve the throwing-catching tasks. Particularly, it can catch the flying objects even on the most challenging catching task with the hands facing sideward.

- We propose a hybrid advantage estimation method, which considers Lyapunov stability. While the effectiveness of the stability condition has primarily been verified for robustness, this is the first time it has been observed to benefit the generation of stable catching behavior for flying objects, thereby improving the accuracy of throwing-catching.

- By using compressed point cloud features of objects as input and applying domain randomization during training, our method excels in generalizing to daily objects with diverse properties (e.g., mass distribution, friction, shape, and initial pose), and performs effectively even on previously unseen objects.

## 2 Related Work

The versatile dexterous hand plays a crucial role in performing various tasks across different human-focused settings. However, effectively managing dexterous hand systems poses a challenge due to their complex actions and intricate contact interactions [15]. Traditionally, controllers for manipulation tasks heavily rely on dynamic models and trajectory optimization techniques [1, 2, 3, 4]. The paper "Catching Objects in Flight" is highly regarded in this field [14]. To enhance model accuracy, their method learns the dynamics of a moving object and develops a reachable and graspable model using concise demonstrations. Furthermore, Williams *et al.* [16] achieved success using the Model Prediction Path Integration Control (MPPI) method to skillfully manipulate a cube. Charles Voss *et al.* [13] improved the MPPI method to manage task handovers between two hands. However, they focused only on a single object due to the complexities involved in model learning.

In recent times, RL-based approaches simplify the controller design process, and model-agnostic techniques have gained substantial popularity within the control domain. In the realm of dexterous manipulation, numerous endeavors have showcased remarkable advancements compared to conven-

tional controllers [17, 18, 19]. Entities like OpenAI have harnessed reinforcement learning-based controllers to successfully rearrange blocks and solve Rubik's cubes [7]. To expedite training, a blend of reinforcement learning and imitation learning has been employed to facilitate dexterous hands in acquiring diverse skills such as repositioning objects, utilizing tools, opening doors, and more [5]. Acknowledging the existing method's limited ability to generalize, Chen *et al.* [8] introduced an effective system capable of learning how to reorient a wide array of objects. For the cooperation between two hands, Zakka *et al.* [20] and Chen *et al.* [19] proposed a series of challenging tasks for bi-manual dexterity, like mastering the piano and lifting the pot. Furthermore, a recent work introduced the Population-Based Training (PBT) algorithm in RL learning, leading to solving the two-handed reorientation task [18]. However, most recent works [18, 21] only focus on a single object or a kind of posture of hands. Our method successfully achieves the throwing and catching of diverse objects, especially for the task with hands facing sideward.

## 3 Preliminaries

### 3.1 Reinforcement Learning

The throwing-catching tasks of dexterous hands can be formalized as an infinite horizon Markov Decision Process (MDP), which is defined by a tuple $(S, A, p, R, \rho_0, \gamma)$. Among them, $S \in \mathbb{R}^n$ is the state space, $A \in \mathbb{R}^m$ is the action space, $p : S \times A \times S \to [0, 1]$ is the transition probability distribution, $r : S \times A \to \mathbb{R}$ is the reward, $\gamma \in [0, 1)$ is the discount factor, and $\rho_0 : S \to [0, 1]$ is the initial state distribution. The goal of RL is to find an optimal cumulative return of the MDP with the policy $\pi : S \times A \to [0, 1]$. The problem is formulated as maximizing the expected discounted cumulative return: $\mathbb{E}_{s_0 \sim \rho_0, a_t \sim \pi(\cdot|s_t), s_{t+1} \sim p(\cdot|s_t, a_t)}[\sum_{t=0}^{\infty} \gamma^t r(s_t, a_t)]$.

### 3.2 Proximal Policy Optimization(PPO)

The PPO algorithm is an on-policy reinforcement learning approach. It excels in robotics tasks featuring high-dimensional action spaces. The algorithm contains an actor and a critic network, where $\theta$ and $\phi$ denote the parameters of the actor network $\pi_\theta(a_t|s_t)$ and the critic network $V_\phi(s_t)$, respectively. For the policy function $\pi_\theta(a_t|s_t)$, the output corresponds to the mean and standard deviation of the Gaussian distribution. On the other hand, the value function $V_\phi(s_t)$ yields the current state's value. Building upon the policy gradient method, the parameter updating formula for the actor-network takes the following form: $\theta_{k+1} = \arg\max_\theta \mathbb{E}_{s_t, a_t \sim \pi_{\theta_k}} [L(s_t, a_t, \theta_k, \theta)]$,

$$L(\cdot) = \min\left\{ \frac{\pi_\theta(a_t|s_t)}{\pi_{\theta_k}(a_t|s_t)} A^{\pi_{\theta_k}}(s_t, a_t), \quad \text{clip}\left[\frac{\pi_\theta(a_t|s_t)}{\pi_{\theta_k}(a_t|s_t)}, 1 - \epsilon, 1 + \epsilon\right] A^{\pi_{\theta_k}}(s_t, a_t) \right\} \quad (1)$$

where $\epsilon$ is a hyperparameter, $A^{\pi_{\theta_k}}(s_t, a_t) = G_t^{\pi_{\theta_k}} - V_\phi^{\pi_{\theta_k}}(s_t)$, is the advantage function. $G_t^{\pi_{\theta_k}}$ is cumulative returns. For parameters $\phi$, the update is expressed as:

$$\phi_{k+1} = \arg\min_\phi \mathbb{E}_{s_t, a_t \sim \pi_{\theta_k}} [(G_t^{\pi_{\theta_k}} - V_\phi^{\pi_{\theta_k}}(s_t))^2] \quad (2)$$

However, because the calculation of $G_t^{\pi_{\theta_k}}$ value needs the whole data of a finished episode, we approximate $G_t^{\pi_{\theta_k}}$ to $r_t + \gamma V_\phi^{\pi_{\theta_k}}(s_{t+1})$ through TD learning method.

## 4 Dexterous Catching Tasks

Our research centers on the challenges of throwing-catching tasks using dexterous shadow hands. Our aim is for these shadow hands to adeptly handle diverse objects in various positions and execute agile catches using different gestures. To address this objective, our design comprises three key components.

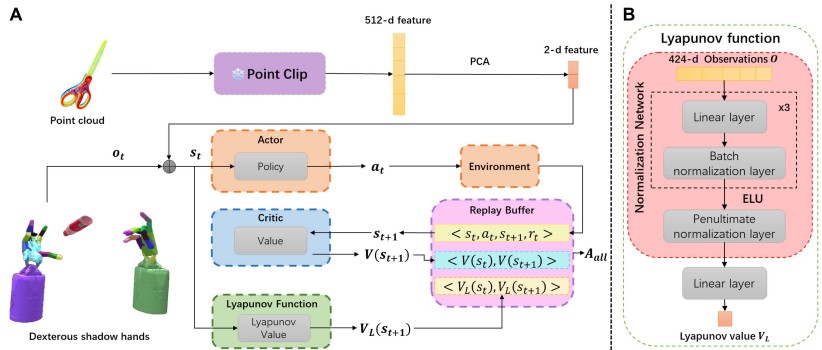

Figure 2: **A**: The method takes as input the environmental observation and the point cloud feature of the object. Then, it learns the catching policy for dexterous hands through an Actor-Critic structure. **B**: The Lyapunov function, the policy function and the value function are estimated using neural networks. Their network structures are similar, and the only difference is the dimension of the output. For example, the Lyapunov function contains a linear layer, a batch normalization layer, and a penultimate normalization layer, and the last linear layer takes a scalar as output.

**Environment Setting**   The shadow hand models, which emulate human skeletons, possess 24 degrees of freedom (DOFs) and are operated using 20 pairs of agonist-antagonist tendons [8]. The presence of force sensors on the ten fingertips facilitates the acquisition of force-related data during catching and throwing maneuvers. In our paper, we investigate five catching scenarios, namely Overarm Catch, Overarm2Abreast Catch, Under2Overarm Catch, Abreast Catch, Underarm Catch and which are shown in Fig. 4 respectively. In these five tasks, one hand should throw an object to the other hand, and the other one should hold it in the palm later. Some tasks with vertical hands are challenging. It is because the object in the hands is not stable without the support from the palm.

**Object-generalized Setting**   Our design objective is to generalize across diverse objects by introducing point cloud features, which are obtained from a pre-trained model, PointClipV2 [22]. Additionally, we apply Principal Component Analysis (PCA) to compress the features. The experiments illustrate that the shapes of different objects can be distinguished using the compressed features. More details can be found in Appendix B.

**State, Action and Reward**   **1) State**: The state space consists of 424 elements. Among these, the 422-dimensional state incorporates crucial information: hand position, hand orientation, angular and linear velocities of the hands, position and speed of each hand's degree of freedom, force-torque sensor readings of the hands, target object's position and orientation, as well as the current object's position and orientation. The above observation corresponds to the setting in [19]. Additionally, a two-dimensional state features the input derived from the point cloud data of the object. It serves as the perceptual part for the agent regarding the object. **2) Action**: The action space comprises 52 dimensions, encompassing 20 pairs of agonist-antagonist tendons for both hands, along with three dimensions each for the required for ce and torque of each hand. **3) Reward**: The reward design comprises three components: the difference in position and posture between the object and the target, the penalty for the action scale of the shadow hand joints, and a combination of task completion bonuses and failure costs. We explain the details of the reward in Appendix C.

## 5   Method

In this section, we introduce the LTC algorithm in detail. LTC is an end-to-end framework based on the Proximal Policy Optimization (PPO) [23] algorithm. Since the throwing-catching task is a dynamic and agile behaviour, the basic PPO algorithm encounters two key challenges: *How to achieve efficient throwing and stable catching without falling?* We make some algorithmic modifications to the basic PPO algorithm. First, to accelerate the learning of throwing, we design an intrinsic advan-

tage and add penultimate normalization layers in the network. Secondly, a stable catching behaviour is holding the object in the palm without falling the object. As depicted in Fig. 3, the maximum sum reward (optimality) can not guarantee the generation of the stable catching behaviour. To encourage more stable catching, we include the Lyapunov stability condition in policy learning. The specific formulation can be found in Section 5.2.1.

## 5.1 Framework

The architecture is illustrated in Fig. 2. Here's a breakdown of its components:1) Point Cloud Feature Extraction: The initial step involves extracting point cloud features using the Point-ClipV2 [22]. Subsequently, Principal Component Analysis (PCA) is employed to reduce dimensionality, yielding a 2-dimensional feature output. This output serves as the perception state for the shadow hands system; 2) Observation Input: The agent takes both the current observation of objects and the shadow hands and the compressed features from point clouds as input; 3) Actor-Critic Framework: The actor and critic are represented as two neural networks, which is the same as PPO algorithm. The network architecture for the Lyapunov function is depicted on the right side of Fig. 2. This structure comprises three linear layers, with a Batch normalization layer and an ELU activation function appended to the output of each layer. Following these, there is a penul-

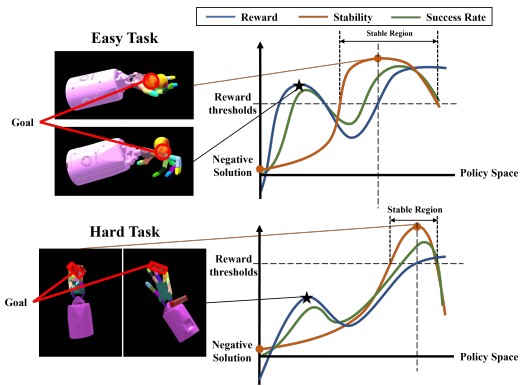

Figure 3: To enhance the success rate, our approach prioritizes optimizing rewards while ensuring the stability of the grasping gesture. By integrating Lyapunov stability alongside reward optimization, we mitigate the risk of completing tasks in unstable poses. The size of the stable region varies based on task difficulty, with simpler tasks offering a larger stable region.

timate normalization layer [24], succeeded by a linear layer to regulate the output dimension. The network structure for the policy and value function are similar to that of the Lyapunov value network.

## 5.2 Hybrid Advantage Estimation

The hybrid advantage $A_{all}$ consists of 3 parts: standard advantage estimation based on PPO, stability advantage estimation, and intrinsic advantage estimation. The formula for the hybrid advantage is expressed as follows:

$$A_{all}(s_t, a_t, s_{t+1}) = \beta_1 A^{\pi_\theta}(s_t, a_t) + \beta_2 A_L(s_t, s_{t+1}) + \beta_3 A_I(s_t, a_t) \tag{3}$$

$$\beta_1 + \beta_2 + \beta_3 = 1 \tag{4}$$

where $A^{\pi_\theta}(s_t, a_t)$ refers to the time differential advantage function of the PPO algorithm, $A_L(s_t, a_t)$ represents the Lyapunov stability advantage function, $A_I(s_t, s_{t+1})$ represents the intrinsic advantage function, and $\beta_1, \beta_2, \beta_3 \in [0, 1]$ are hyperparameters. As follows, we introduce the Lyapunov stability advantage and intrinsic advantage in detail.

### 5.2.1 Lyapunov Stability

The Lyapunov function serves as a potent tool for assessing system stability. One approach, known as the self-learning Lyapunov stability method [25], is utilized in our paper. First, we introduce a Lyapunov stability critic network, which represents the candidate Lyapunov function $V_L$ using neural networks. Subsequently, the empirical Lyapunov risk $\mathcal{R}(\Lambda)$ [25] is defined as per Eq. (5). The three parts in Eq. (5) represent three requirements for the Lyapunov function [25]. The parameter update is then executed by minimizing the loss function associated with the Lyapunov risk $\mathcal{R}(\Lambda)$, as part of achieving the desired system stability.

$$\mathcal{R}(\Lambda) = \mathbb{E}[\max(-V_L(s_t), 0) + \max(0, L_f V_L(s_t)) + V_L^2(0)] \tag{5}$$

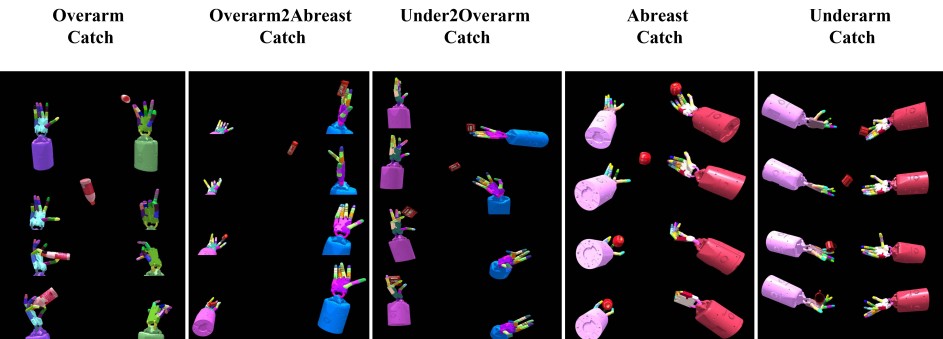

Figure 4: Video snapshots of throwing-catching tasks performed by our method. There exist five tasks with different postures of hands, in order from difficult to easy: Overarm Catch, Overarm2Abreast Catch, Under2Overarm Catch, Abreast Catch, and Underarm Catch. The most challenging task is Overarm Catch because the hands are both oriented vertically. Notably, our method can catch diverse objects in five tasks.

where $f$ is the dynamical system, $s_t$ is the state at step $t$. The empirical Lyapunov risk is non-negative, we approximate the Lie derivative $L_f V_L(s_t)$ by finite difference of the system's sampling trajectory because of the complexity of modeling this system's dynamics:

$$L_f V_L(s_t) = \frac{1}{\Delta t}(V_L(s_{t+1}) - V_L(s_t)) \tag{6}$$

where $\Delta t$ is the time difference between them. Since the Lyapunov function is considered to be another value function, the Lie derivative term $L_f V_L(s_t)$ is the only term in the Lyapunov risk affected by the policy. Therefore, an additional goal of ensuring $L_f V_L(s_t) < 0$ is added to the policy optimization, to encourage the policy's update with the requirement of stabilization. However, the classical constraint $L_f V_L(s_t) < 0$ makes the value of the Lyapunov function likely to be stuck in a minimal number close to 0. This drawback hinders the process of policy improvement due to the lack of an explicit gradient.

Therefore, we design a more strict constraint $L_{f_{\pi_\theta}} V_L(s_t) < -k V_L^\alpha(s_t)$, where $\alpha$ and $k$ are constants ranging from 0 to 1. Note that $k V_L^\alpha(s_t)$ is non-negative. When $k V_L^\alpha(s_t)$ is larger, an intuitive phenomenon is that $V_L$ approaches 0 faster. In other words, the system can reach a stable region in fewer steps, which benefits the generation of stable catching behaviors. More details can be found in Appendix D. Hereafter, we design the part of the Lyapunov stability advantage function based on the clipped Lie derivative term:

$$A_L(s_t, a_t) = \min(k V_L^\alpha(s_t), -L_{f_{\pi_\theta}} V_L(s_t)) \tag{7}$$

### 5.2.2 Intrinsic Advantage

Based on the advantage function of the PPO algorithm, in order to improve the ability of the system efficiently, we design the intrinsic advantage function with reference to the intrinsic reward mechanism [26]. For a Markov decision process trajectory, different values $V_\phi(s_t)$ will affect the success of the final result of this trajectory. To provide a specific example, let's consider the fact that the next states $s_{t+1}$ exhibit varying conditions, leading to different values $V_\phi(s_{t+1})$. It's worth noting that in a successful process, the value $V_\phi(s_{t+1})$ tends to be higher. Consequently, within a Markov decision process, we can further refine the advantage function $A(s, a)$ using empirical measures. If an action is effective, it should result in a higher value, as represented by $V_\phi(s_{t+1}) - V_\phi(s_t) > 0$. This signifies that action $a_t$ transitions to a state with a higher success rate. Conversely, an unfavorable action can lead to a decrease in value, reflected as $V_\phi(s_{t+1}) - V_\phi(s_t) < 0$. In order to reduce failure in the early stage of the throwing process, we consider the intrinsic advantage $A_I(s_t, s_{t+1})$ as part of the total advantage function to accelerate the RL training, shown as follows:

$$A_I(s_t, s_{t+1}) = \min(k(V_\phi(s_{t+1}) - V_\phi(s_t)), 0) \tag{8}$$

where $k$ is a positive constant.

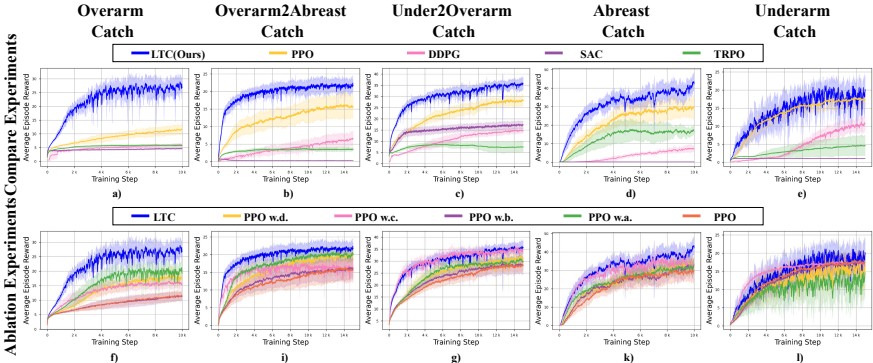

Figure 5: The average reward curve for training multiple objects for five different tasks. The first row is the comparison experiment reward graph. The second row is an ablation experiment reward graph. (Curves smoothed report the mean 5 times, and shadow areas show the variance.)

# 6 Experiments

To test the algorithm's ability to generalize, we train and test it with objects from the YCB_Pybullet dataset [27]. The results show that our algorithm outperforms baseline methods. Four essential components of our algorithm improve its performance. Additionally, our method showcases the ability to generalize different gestures across five skilful throwing-catching tasks, effectively allowing for the successful manipulation of various objects. Importantly, our approach extends its capabilities to catch unseen objects successfully.

## 6.1 Comparison Experiments

In this section, we conduct a comparative analysis of our algorithm against mainstream reinforcement learning (RL) algorithms in dexterous hands' throwing-catching tasks. These algorithms include TRPO [28], SAC [29], DDPG [30], and PPO [23]. The results, as illustrated in Fig. 5, are quite telling. In the most challenging task, the Overarm Catch, our algorithm significantly outperforms the baseline methods. For the Over2abreast Catch, Under2Overarm Catch, and Abreast Catch tasks, our approach demonstrates superior learning efficiency in acquiring throwing and catching skills. Furthermore, even in the relatively easier task, Underarm Catch, our method exhibits slightly improved learning efficiency and performance compared to the baseline algorithms. However, the variance of the training curve is larger than those baselines in Underarm Catch. This is because simpler tasks have more gestures to complete tasks (see Fig. 3), leading to numerous feasible solutions. Thus, the strong exploration of our method results in a high variance of the sum of rewards. Conversely, complex tasks feature limited gestures, resulting in minimal reward variance post-completion. Video snapshots illustrating the performance in five tasks can be seen in Fig. 4. It's worth emphasizing that we train the model using nine objects with diverse shapes and mass distributions. This showcases the algorithm's robust generalization ability, as it can adapt to various objects and perform tasks efficiently and reliably.

## 6.2 Ablation Experiments

We evaluate the impact of four crucial components in our algorithm. To clarify, here are their definitions: **a.** indicates the addition of a Lyapunov stability component. **b.** indicates the addition of an intrinsic advantage component. **c.** indicates network structure parameter normalization, and **d.** indicates that the compressed point cloud feature of the object is added as a prior to the state. The effect of these components is shown in the reward curve in Fig. 5. It can be seen from the figure

Table 1: Success rate of Task

| Task | Success Rate (%) | | | | | | | | |
|---|---|---|---|---|---|---|---|---|---|
| | Train (9 objects) | | | | Test (11 objects) | | | | |
| | **META** | Apple(regular) | Banana(elongated) | Poker(flat) | **META** | Stapler | Scissor | Washer | Bowl |
| Overarm | **60.30** | 63.62 | 73.51 | 73.53 | **67.58** | 69.71 | 56.83 | 76.00 | 77.34 |
| Overarm2Abreast | **68.77** | 79.01 | 82.89 | 68.76 | **76.79** | 81.24 | 83.90 | 67.42 | 86.96 |
| Under2Overarm | **73.95** | 75.35 | 82.21 | 78.79 | **73.41** | 76.85 | 68.46 | 57.34 | 81.88 |
| Abreast | **72.95** | 67.41 | 88.65 | 86.34 | **77.89** | 94.84 | 57.52 | 83.94 | 83.59 |
| Underarm | **88.99** | 95.87 | 86.50 | 90.57 | **80.47** | 84.10 | 63.63 | 63.10 | 83.23 |

[1] After dividing the data into training and test sets, we perform distinct success rate tests(10k times) for single objects and multiple objects. Specifically, we only display the success rate of three typical objects within the training set and four typical objects within the test set. We utilize a reward threshold to ascertain the success rate. In the Under2Overarm Catch task, success is determined if the reward surpasses 20 because of the longer execution process, while for other tasks, a reward exceeding 15 indicates success.

that each component of our method benefits the performance compared with the PPO algorithm. The inclusion of Lyapunov stability (**a.**) in the baseline PPO notably enhances the hand's catching ability, particularly accentuated during complex tasks like the Overarm Catch. There exists a more detailed explanation of ablation experiments in Appendix E.

## 6.3 Generalization Ability

We evaluate our method in the throwing-catching task using various objects, quantifying the success rate as a key metric. We run the testing task in 10k episodes and use the average reward threshold to determine whether it is successful. Since our algorithm pays more attention to the efficiency and stability of generalization ability learning, we no longer use the distance threshold of the final state from the target position as the success rate. The success rate of different objects is illustrated in the Table 1. Among them, we use 11 unseen objects (test set) to verify the generalization ability of our method. Specifically, we adopt two methods to test objects in the training and test sets: i. Test the success rate of multiple objects in parallel. ii. Test the success rate of each object individually. We take the most challenging task, Overarm Catch as an example. The average success rate for the training set objects reaches 60.30%, and the highest of individual tests on each object was pen up to 81.23%. In addition, the average success rate of the test set object is close to 67.58%. Most importantly, our method obtains a good success rate, 56.83%, in testing scissors, which is a new but difficult object. Furthermore, we can observe that our method can provide a robust catching strategy because there is not a significant gap between the success rates of training and test objects. The results provide strong evidence of our algorithm's robust generalization capability for catching diverse objects. We perform a more detailed analysis of the generalization ability in Appendix E.

## 7   Conclusion

**Summary**: In this study, we introduce an end-to-end framework (LTC) based on model-free reinforcement learning to tackle throwing-catching tasks with dexterous hands. Remarkably, our method demonstrates exceptional generalization ability, even for unseen objects, and effectively handles scenarios where the hands face sideways or other postures. This research underscores the potential of RL-based approaches in improving object transportation efficiency through dynamic manipulation with dexterous hands.

**Limitation**: Addressing the randomness for more complex objects or the initial position and posture of the hands is a key challenge for our method. To enhance robustness and success rate, we plan to iteratively refine task randomness by combining curriculum learning methods. Furthermore, we intend to optimize the motion and dynamic control parameters through adaptive parameter learning methods. Additionally, we've constructed a robot hardware platform comprising a mobile chassis, a 7-DoF robotic arm, and an allegro hand. Through the meticulous design of domain randomization terms, we hope to achieve zero-shot simulation-to-real transfer and execute throwing-catching tasks in real-world settings.

**Acknowledgments**

Thank you to all the authors and those who put in the effort in this work. This work was supported by the National Natural Science Foundation of China [grant numbers U21B6002]. This work is also supported by the Ministry of Science and Technology of the People´s Republic of China, the 2030 Innovation Megaprojects "Program on New Generation Artificial Intelligence" (Grant No. 2021AAA0150000) and the National Key RD Program of China (2022ZD0161700).

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

# A  Implementation detail

In our experiments, we conduct training and testing on a platform equipped with an RTX A6000 GPU and an AMD CPU. We utilize nine objects for training across five different task types and test the algorithm with four objects. Our approach also leverages meta-learning for multi-object parallel training. The objects used in our experiments are sourced from the YCB dataset [27]. Due to the limitations of the platform, we employ 64 parallel environments per object for the Abreast Catch task and 256 parallel environments for others. The policy updates every 8 environment steps, and the length of each episode is set to 100 steps. Furthermore, every experiment is built using early reset functions mentioned in Appendix B. To determine the reward threshold, we randomly sampled 50 experimental environments during the testing phase. By visualizing and analyzing the corresponding reward values after successfully catching the objects, we calculated the average and rounded it to establish the reward threshold.

# B  Object-generalized design

The 3D point cloud feature of an object represents its shape, created by densely gathered points in 3D space. We adopt point clouds to approximate object features as prior information. Our approach involves feature extraction via a point clouds processing network utilizing PointClipV2 [22]. This algorithm facilitates zero-shot classification segmentation to identify features. To reduce computational complexity, the extracted feature map degrades to two dimensions using Principal Component Analysis (PCA). Our visual analysis has substantiated that these 2-dimensional features adequately capture the distinct attributes of various objects, as demonstrated in Fig.6. The distribution of various objects exhibits three distinct directions, leading to their classification into three categories. Elongated objects, like bottles, fall into the first category. Flat objects,

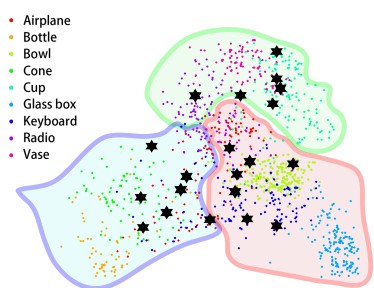

Figure 6: The features of different point clouds are extracted by PointClipV2 [22], then reduced by PCA, and finally visualized in two dimensions. The Clustering Using Representatives(CURE) method shows that any object is approximately divided into three categories according to different shapes, extending in three directions, respectively, elongated, flat, and regular. Sample of objects in each category are bottles, keyboards, and cups, respectively.

such as keyboards, form the second category. Regular objects, like cups, which do not fit into the previous categories, are classified under the third category.

Furthermore, to enhance the algorithm's ability to generalize across various physical properties of objects, we broaden the training dataset by randomly generating mass and inertia values within a reasonable range for the same object. This approach increases the diversity of physical properties encountered during training. In addition, we reset the environment when the object falls under a threshold or gets out of some bounds. The process enhances the diversity of effective data in the early stage of training.

During the training, we leverage the IsaacGym simulator [31], which enables parallel training, allowing the system to learn the skill of catching various objects simultaneously.

# C  Reward design

The reward is composed of five elements: 1) $r_p$ quantifies the distance between the position of the goal object $p_g$ and the current object $p_c$. 2) $r_o$ represents the angular discrepancy between the target object's orientation and the current object's orientation, utilizing quaternions for accurate angular measurement. 3) $r_a$ evaluates the smoothness of joint movement through the current

actions $a_c$. 4) A constant reward $c_1$ is given for successfully completing tasks $r_{succ}$. 5) A constant penalty $c_2$ is imposed for task failure $r_{fall}$. Specifically, the smoothness of joint movement is controlled by negatively rewarding the sum of squares of action quantities as per the strategy. The overall reward is expressed $r_{total}$ in Table 2, where target position distance is calculated using Euler distance, angular distance is determined using quaternions, joint smoothness is penalized using action quantities' sum of squares, and success and failure rewards are assigned as positive and negative constants, as outlined in Table 2.

Table 2: Reward Desgin

| | |
|---|---|
| $r_p$ | $\|p_g - p_c\|_2$ |
| $r_o$ | $2\arcsin(\frac{\|q_d\|_2 - \|q_d\|_{min}}{\|q_d\|_{max} - \|q_d\|_{min}})$ |
| $r_a$ | $-\sum a_c^2$ |
| $r_{succ}$ | $c_1$ |
| $r_{fall}$ | $c_2$ |
| $r_{total}$ | $r_p + r_o + r_a + r_{succ} - r_{fall}$ |

[1] The reward is composed of five parts, involving the position, posture of the object, task success or failure, and smooth actions.

## D Details of Lyapunov Stability

We design a stricter constraint: $L_{f_{\pi_\theta}} V_L(s_t) < -kV_L^\alpha(s_t)$, where $\alpha$ and $k$ are constants ranging from 0 to 1. Note that $kV_L^\alpha(s_t)$ is non-negative. This constraint avoids the issue of the Lyapunov function having very small values, thus speeding up the convergence of the stability part in policy improvement.

Our idea is inspired by a classical theorem on finite-time stability convergence in a continuous-time system [32].

**Lemma D.1** (Finite-Time Stability Convergence). *In a continuous-time system, the system can be stable within a finite time* $T \leq \frac{V_L^{1-\alpha}(s_0)}{k(1-\alpha)}$, *if the following condition holds.*

$$\dot{V}_L \leq -kV_L^\alpha \tag{9}$$

*Note that $s_0$ is the initial state, $\dot{V}_L$ is the Lie derivative $L_f V_L$, $k$ and $\alpha$ are constants ranging from 0 to 1.*

*Proof.* First, we perform an integral operation on both sides of the condition.

$$\int \frac{\dot{V}_L}{V_L^\alpha} dt \leq \int -k dt \tag{10}$$

Then, we can obtain

$$\frac{1}{\alpha - 1} \frac{-1}{V_L^{\alpha-1}} \big|_0^T \leq -kt \big|_0^T$$

$$\frac{1}{\alpha - 1}(V_L^{1-\alpha}(s_0)) \leq -kT \tag{11}$$

$$\frac{V_L^{1-\alpha}(s_0)}{1-\alpha} \geq kT$$

where $s_0$ is the initial state. Finally, the convergence time $T$ can be represented as:

$$T \leq \frac{V_L^{1-\alpha}(s_0)}{k(1-\alpha)} \tag{12}$$

□

Compared with $\dot{V}_L \leq 0$, the following lemma accelerates the convergence time from infinity to finite time. We can provide an intuitive explanation. Equation 9 guarantees that the derivative of the Lyapunov function $V_L$ is constantly less than a large negative value $(-kV_L^\alpha)$ instead of 0. Thus, we can similarly use the condition of Equation 9 in our method. This process not only facilitates the policy in finding behaviors with better stability properties but also avoids the values of the Lyapunov function being close to 0 in the entire state space. For the choice of $\alpha$ and $k$, we set $\alpha$ to 0.7 and increase $k$ linearly from 0.1 to 1 according to the training steps. The small $k$ in the early stage can alleviate the collapse of the model.

# E    Ablation Experiment Analysis

The inclusion of Lyapunov stability ($a$.) in the baseline PPO notably enhances the hand's catching ability, particularly accentuated during complex tasks like the Overarm Catch. By guiding dexterous hands to avoid unstable gestures and focusing on learning stable catching motions, we achieve higher success rates and rewards. It is worth noting that in the simplest task, the Underarm Catch, the effect of Lyapunov stability is limited. As seen in Fig.3, the task's stability region is large, thereby making it hard to balance between stability and rewards. Compared with PPO, it is more challenging to explore optimal solutions for success rate.

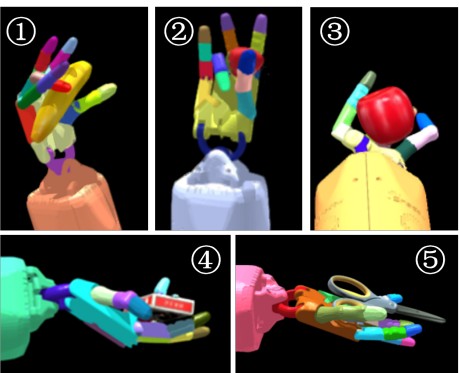

The impact of increasing intrinsic advantage ($b$.) is observed primarily during the early stages of learning. While its effect may not be substantial, it demonstrates the capability to expedite early learning and convergence across multiple tasks. Parameter normalization ($c$.) significantly enhanced the ability of the throwing process. The normalization layer accelerates the early convergence of the model due to a more stable gradient. The stable gradient update benefits searching the optimal point, particularly in tasks with relatively simple throwing processes. It performs well in tasks where the success of the complete throwing-catching task depends heavily on the throwing process, such as Under2Overarm Catch, Underarm Catch, and Abreast Catch. The incorporation of the object's point cloud feature ($d$.) simultaneously considers both the throwing and catching processes, offering valuable cues for each. This enhancement notably boosts the al-

Figure 7: For the regular, flat, and elongated classes of objects, the model generalizes three different stable catching strategies for unseen objects in each of the three tasks. For regular objects such as apples and strawberries, it uses half-grip (③) or pinching gestures (②). A full-grip gesture (①) is used for slender objects such as bananas. For flat objects such as scissors and pokers, the hand can hold them directly in the palm (④,⑤).

gorithm's performance by improving convergence speed and task success rates. Given that the object's shape affects the catching, this feature significantly improves tasks with challenging catching behavior like Overarm Catch. Because the throwing-catching behavior is insensitive to the object's shape in Underarm Catch, adding point cloud to extend state space harms the performance slightly.

# F    Generalization Ability

**For diverse objects:** In this section, we delve into a detailed analysis of our method's generalization capabilities. It demonstrates the ability to execute distinct catching gestures based on the positions of the shadow hands and the objects being caught. Through an analysis of the high-dimensional point cloud features, we categorize most objects into three primary classes. Remarkably, our method exhibits over three different catching gestures for each of these three object types, as depicted in Fig. 7. For objects falling into the regular, flat, and elongated categories, the model generalizes three distinct

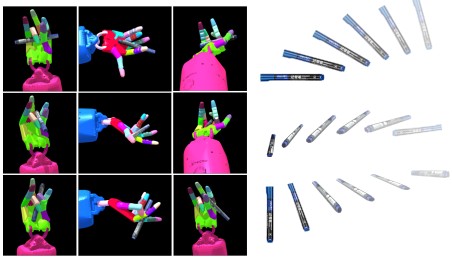

Figure 8: When the pen is oriented obliquely or tumbling because of the randomness in the initial posture, the shadow hand can generalize various grasping gestures to successfully complete the tasks.

gestures in each of the five tasks. For instance, when dealing with regular objects like apples, it employs either a half-grip or pinching gesture. Slender objects like bananas are handled with a full-grip position. Flat objects, such as scissors and pokers, are directly held and secured between the fingers

and palms. This versatility in catching gestures reflects the algorithm's remarkable adaptability to different object types and positions.

**For different initial postures:** Furthermore, given that the initial position of the object is randomly selected in each episode, the object's motion follows different trajectories due to aerodynamic processes. Impressively, our method excels at solving dynamic catching tasks by generating diverse catching gestures, as illustrated in Fig. 8. For example, when capturing a pen, we observe that the catching hand can adapt to grasp the pen with varying postures. In practical scenarios, catching a randomly rotating pen in mid-air, especially due to its anisotropic shape, would be challenging. However, our results demonstrate the algorithm's ability to catch randomly rotating objects in the air, even when dealing with challenging objects like pens.

## G   Simulation to Real World Design

To ensure the successful transfer from the simulation environment to the real world, we address key challenges associated with sim-to-real transfer and offer specific solutions to overcome them.

*Challenge 1: Can the observations obtained during the simulation process be effectively acquired in the real world?*

**Solution 1**: In the following, we introduce how to obtain each component of the observation in the real world.

- Position, rotation, velocity, target position, rotation of the object. We have validated the use of RGB image input from the main viewing angle, utilizing GPT-4V or other multimodal large models in combination with object detection and segmentation technologies. This approach enables us to obtain the initial pixel position of both the current object and the target. Real-time 6D pose estimation allows us to track the object's pose and pixel position during its flight. By integrating data from a binocular or depth camera, we can extract the object's three-dimensional world coordinates, and by leveraging the video stream frame rate, we can calculate the object's motion speed. We test using the FAIRINO ROBOT 3 robotic arm and the RealSense D435 camera on our platform. The execution frequency for detection, segmentation, and 6D pose estimation reached 10Hz, demonstrating that our system meets the initial performance requirements for executing dynamic tasks.

- Point cloud prior information of the object. The object point cloud feature serves as prior information, capturing the shape characteristics after standardizing the object's size. This feature does not need to be acquired in real-time during the process. When acquisition is not feasible, the point cloud feature can be substituted with the representative geometry of the three types of objects detailed in Appendix B.

- The position, rotation, and speed of shadow hands' wrists and joints. These can be obtained from the robot itself.

- The force of the fingertips. These can be obtained through a fingertip force sensor.

- The object's physical properties and the hand as well as the release velocity of the object being thrown. The terminal motion velocity can indeed be derived from the joint angular velocity, specifically the wrist movement velocity. The initial velocity of the object can be estimated through visual recognition, and subsequently corrected by combining the terminal motion velocity with six-dimensional force/torque data from the wrist sensors.

*Challenge 2: How can we address the observation errors that occur in the real-world observations?*

**Solution 2**: To account for potential errors in real-world observation acquisition, we introduce a pose perturbation of ±5° on each of the three axes of the current hand posture and a perturbation of ±5 cm in the distance between the hands. This approach is designed to test the policy's tolerance for variations in hand positions and postures. Additionally, we incorporate an observation error of ±5 cm and a pose perturbation of ±20° on each axis for the position and pose measurements of

the object and target, respectively, along with an observation error of ±20% for the object's velocity measurement. The success rate measured after applying all these errors is compared with the original success rate, as presented in Table 3.

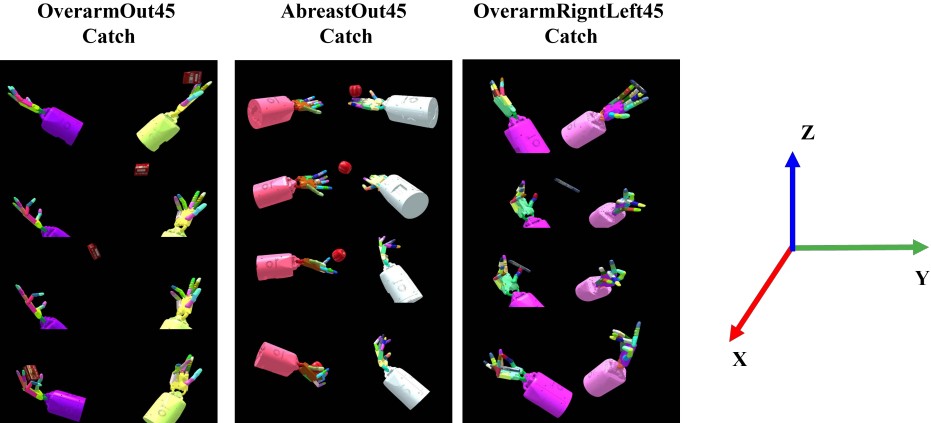

Figure 9: Video snapshots of throwing-catching tasks performed by our method. The description of tasks can be found in Appendix G.1.

Table 3: Success rate of Task with disturbance

| Task | Success Rate (%) | | |
|---|---|---|---|
| | Orignal(11 object) | | Add disturbance (6 object) |
| | META(Train) | META(Test) | META |
| Overarm | 60.30 | 69.97 | 60.17 |
| Overarm2Abreast | 68.77 | 79.88 | 66.75 |
| Under2Overarm | 73.95 | 71.06 | 70.09 |
| Abreast | 72.95 | 79.97 | 68.84 |
| Underarm | 88.99 | 73.51 | 83.65 |

*Challenge 3: Can the action outputs from the simulation process be implemented for control in the real world?*

**Solution 3**: To ensure that the actions generated in the simulation can be effectively controlled in the real world, our algorithm framework utilizes most position-based control, and minimizes reliance on force-based control. In our approach, the finger joints of the shadow hands are position-controlled, while only the force and torque of the wrists are controlled. When the robotic arm is connected to the wrist and transitioned to the real-world setting, the planning and control of the robotic arm can be handled by an impedance controller after getting the force and torque of the wrists. This design facilitates future research and verification tasks involving the connection of the wrists to the robotic arm.

*Challenge 4: Can our method generalize to different hand poses with significant variations?*

**Solution 4**: Although we believe that the five hand postures designed in this work already include the main combinations of the three axes of the two-hand xyz, and the other hand postures can be varied from these hand postures, in order to ensure the effectiveness of our policy, we added three more general postures: overarmout45, abreastin45, and overarmrightleft45, and achieved a good meta-success rate on the test set and the training set after training, as shown in Table 4. A schematic diagram of the task is shown in Figure 9.

Finally, regarding future work, we will support further research on the combination of dexterous hands with robotic arms.

Table 4: Success rate of New Task

| Task | Success Rate (%) |
|------|------------------|
|      | **META** |
| Overarmout45 | **61.44** |
| Abreastin45 | **68.77** |
| Overarmrightleft45 | **85.41** |

### G.1 Description of Added Tasks

The new tasks are depicted as follows.

- **Overarmout45 Catch**: The hands are oriented vertically. Then, the throwing hand rotates around the x-axis of the reference coordinate system (shown on the right side of Fig. 9) by -45 degrees, while the catching hand rotates around the x-axis by 45 degrees.

- **Abreastin45 Catch**: The setting of hands is built upon the Abreast Catch task. Then, the throwing hand rotates around the z-axis of the reference coordinate system (shown on the right side of Fig. 9) by 45 degrees, while the catching hand rotates around the z-axis by -45 degrees.

- **Overarmrightleft45 Catch**: The hands are oriented vertically. Hereafter, the throwing hand rotates around the y-axis of the reference coordinate system (shown on the right side of Fig. 9) by 45 degrees, while the catching hand rotates around the y-axis by -45 degrees.

## H Comparison Experiments Additional Notes

To further demonstrate the effectiveness of the proposed method compared to baseline and other reinforcement learning approaches, we conducted an additional comparative experiment, focusing on the success rate, as shown in Table 5. The results clearly indicate that our approach consistently outperforms both the baseline and other methods across all metrics.

Table 5: Success rate of Task with Other Algorithm

| Task | Success Rate (%) | | | | | | | | | |
|------|------|------|------|------|------|------|------|------|------|------|
|      | LTC | | PPO | | TRPO | | SAC | | DDPG | |
|      | META(Train) | META(Test) | META(Train) | META(Test) | META(Train) | META(Test) | META(Train) | META(Test) | META(Train) | META(Test) |
| Overarm | 60.30 | 69.97 | 5.16 | 0.71 | 0.84 | 0.96 | 0.00 | 0.39 | 0.58 | 0.65 |
| Overarm2Abreast | 68.77 | 79.88 | 24.75 | 34.41 | 0.00 | 0.54 | 0.00 | 0.00 | 6.81 | 5.51 |
| Under2Overarm | 73.95 | 71.06 | 62.76 | 66.34 | 14.88 | 19.12 | 68.82 | 69.86 | 53.85 | 59.24 |
| Abreast | 72.95 | 79.97 | 43.08 | 52.76 | 47.91 | 44.88 | 0.00 | 0.00 | 0.00 | 0.00 |
| Underarm | 88.99 | 73.51 | 70.74 | 65.81 | 4.01 | 4.425 | 0.00 | 0.00 | 0.00 | 0.00 |

