# OpenReview forum: "DexCatch:  Learning to Catch Arbitrary Objects with Dexterous Hands"
_robot-learning.org/CoRL/2024/Conference — CoRL 2024_

### Official Review · Reviewer_RD6F · 2024-07-17

**Originality:** 3
**Technical Quality:** 3
**Clarity Of Presentation:** 3
**Potential Impact:** 3
**Recommendation:** 3
**Confidence:** 4

**Review:**

Strength:
1. The throwing and catching task is dynamic and challenging, and the authors categorize the tasks into 5 different settings, which is clear.
1. The stability function idea is useful for learning stable policies in the RL framework.
1. The paper is clear and the experiments are well-designed.

Weakness:
1. The novelty of the method is limited. It looks like a mix of PPO and Lyapunov functions.
1. There are no real-world results, which I believe matter a lot for RL papers in simulation since anything can happen in simulation and RL can always figure something out. The sim2real gap is large for such dynamic and contact-rich tasks. It would be good if you can show sim2real results or provide a more exhaustive list of sim2real challenges and plans to address them.
1. The quality of the figures and the video can be improved. For example, the text in the figure is too small and the black background makes it hard to tell the robot hand and the object.
1. The throwing motion looks unrealistic, e.g., a 720-degree rotation before releasing the object. Hard for me to imagine this working on a real robot. I understand the necessity of such motion to create momentum, but maybe it's more realistic to connect the hand to a robot arm and use the arm motion to create momentum. Connecting the hand to a robot arm in the simulation also adds additional kinematic constraints to the policy to ensure realistic motions.
1. The success rate for the training set objects is unsatisfying, although you can argue that the task is very difficult.
1. Regarding the plan for real-world experiments, I’m a bit concerned about the embodiment gap between the allegro hand and the shadow hand.

**Quality Of The Limitations Section:**

3

**Questions For Rebuttal:**

1. Does the action space include the wrist pose for both hands?
1. What’s the current and target object pose on line 123? How do you get the target object pose?
1. In Section 6.1, using the mean episode reward as the only metric seems not very satisfying. Can you include e.g. the success rate of the policy trained by different networks like in Table 1?
1. How do you decide the reward threshold for the success rate? “Lines 252-253 in Section 6.3 need more explanation.
1. What does w.d. w.c. w.b. w.a. mean in Figure 5?
1. What would be the main challenges for sim2real transfer? Some things I can think of: How do you acquire the point cloud? Depth cameras in the real world will probably suffer from occluded surfaces, blurred motion, noisy points positions, and features. Will the policy be responsive enough given that it looks like a colored point cloud is needed for the input? Which information is directly available in your state space? How do you estimate the state information for those not directly available?
1. Contact-rich and dynamic tasks like this usually have very different physics in the real world. Given that you’re using IsaacGym, whose physics I’m not very confident about, why do you think you can do zero-shot transfer?

**Robotics Focus:**

3

**Summary Of Paper:**

This paper introduces LTC, a learning-based framework for dexterous throwing-catching tasks. The method incorporates Lyapunov stability for improved catching accuracy and stability. Using compressed point cloud features and domain randomization, LTC generalizes well to diverse objects, including highly challenging scenarios.

**Summary Of Recommendation:**

I recommend a weak reject for this paper. While it presents a clear and well-designed framework for the challenging task of dynamic throwing and catching with promising simulation results, the novelty of the method is limited and lacks real-world validation. The paper also faces issues with unrealistic motions, and lower success rates for training set objects, highlighting concerns about the sim-to-real gap.

---

### Official Review · Reviewer_2Jip · 2024-07-21
**Learning to catch with dexterous hand: novel framework, more experiments recommended**

**Originality:** 3
**Technical Quality:** 3
**Clarity Of Presentation:** 4
**Potential Impact:** 3
**Recommendation:** 3
**Confidence:** 3

**Review:**

Overall, the paper is structured and readable. The proposed framework is novel and relevant to the CoRL audience. The relevant work seems adequate. The proposed framework is clearly presented with details on the advantage functions, making it easier for potential reproductions. The problem formulation and policy training method are generalizable to other dynamic manipulation tasks. The evaluation can be strengthened by comparison of success rate and generalization ability test in domains other than objects and initial hand poses.

Strengths:
- The integration of Lyapunov stability is significant to enhance the performance of reinforcement learning in dynamic tasks.
- The zero-shot transfer to unseen objects, facilitated by compressed point cloud object features and domain randomization during training, demonstrates the robustness of the framework.
- The simulation results show that the proposed framework outperforms the baselines regarding training efficiency and achieves a relatively high success rate.
- The ablation experiments indicate that all components of the proposed method are crucial to the performance of the system.

Weaknesses:
- The generalization ability is not tested regarding the distance between the two hands.
- Dynamic tasks are susceptible to external factors in the real world, such as air drag. The robustness of the proposed framework in real-world tasks is unclear.

**Quality Of The Limitations Section:**

2

**Questions For Rebuttal:**

1. While the experiment results clearly show that the proposed method has better training efficiency compared to the baselines, there does not seem to be a quantified comparison between the success rates of different methods.
2. More settings in the simulation can be changed during experiments to test generalization ability, such as the distance between the hands.

**Robotics Focus:**

3

**Summary Of Paper:**

The authors propose a learning-based framework for throwing-caching tasks using dexterous hands. It is an end-to-end framework that takes as input the environmental observation and the point cloud feature of the object and then learns the policy through an actor-critic structure. The framework employs Proximal Policy Optimization (PPO) algorithm, enhanced with modifications, to address the challenges of dynamic manipulation. Specifically, it integrates standard PPO advantage estimation with Lyapunov stability and intrinsic advantage estimations to improve stability and efficiency in learning. The proposed method is evaluated on various tasks using two shadow hands in simulation, based on training efficiency and success rate of throwing and catching different objects.

**Summary Of Recommendation:**

The authors present a framework that is relevant and novel, they provide detailed information that helps the readers to understand the entire pipeline, from design of the inputs to the implementation of advantage functions during policy training. The framework is well grounded and tested on various objects in the simulation. The authors might further strengthen this paper by running more experiments to prove that the framework is robust and generalizable.

---

### Official Review · Reviewer_JPgA · 2024-07-22
**A Potential Framework for Learning to Solve Throwing-Catching Tasks in Simulation**

**Originality:** 3
**Technical Quality:** 3
**Clarity Of Presentation:** 3
**Potential Impact:** 3
**Recommendation:** 3
**Confidence:** 4

**Review:**

The authors propose a reasonable approach to learning throwing-catching behavior. Tackling dynamic dexterous manipulation tasks is challenging yet important. It’s good to see the authors’ effort in this.
Strength:
- The use of hybrid advantage to successfully train the shadow hand to throw and catch the object in simulation is an important and clever way.
- It is impressive to see the Lyapunov stability benefit the generation of stable catching behavior for flying objects.
Weakness:
- Further discussion is needed regarding the effectiveness of the trained policies in more general hand posture tasks (beyond those hand postures included in the 5 throwing-catching scenarios mentioned in Line111).
- The discussion is insufficient regarding sim2real transfer and challenges that might be encountered. As the state space consists of the information (such as the position of the object, velocities of the object, etc.) that is easily available in the simulator but not in the real world, the method’s strength won’t be adequately established until evaluated against a full transfer to the real word.

**Quality Of The Limitations Section:**

2

**Questions For Rebuttal:**

It is stated in Line 250 that “…use the average reward threshold to determine whether it is successful.” However, in Line415 and Line416, one of the components of the reward function is a constant reward r_succ which is given for successfully completing tasks. It seems that there is a loop that how to determine whether it is successful is based on the reward while the reward is related to whether it is successful. Please explain. And I would like to know whether r_succ is determined by the distance threshold of the final object position from the target position. If it were, what’s the meaning of using the average reward threshold instead of the distance threshold of the final state from the target position as the success rate (mentioned in Lines 251-253). Since the reward is related to the distance threshold, it seems that there is a redundancy as I’m not sure whether or not the average reward threshold provides more information about the task completion than the distance threshold. Please explain.

**Robotics Focus:**

3

**Summary Of Paper:**

The authors present a framework (LTC) based on model-free reinforcement learning for throwing-catching tasks using dexterous hands in simulation. In the proposed framework, an intrinsic advantage, the normalization layers and the Lyapunov stability condition are added to the basic PPO algorithm to train better policies. The learned policies can generalize to unseen objects and effectively handle unstable scenarios.

**Summary Of Recommendation:**

I think the paper presents a nice framework for throwing and catching tasks in simulation. I appreciate the authors’ effort but I feel discussion and clarification regarding the mentioned weaknesses are necessary before acceptance.

---

### Author Rebuttal · Authors · 2024-08-12

The revised manuscript and supplementary video are attached in the file.

---

### Decision · Program_Chairs · 2024-09-05

**Decision:**

Accept

**Comment:**

Update: I have read the author s rebuttal  and my concern still stands that this framework would not be realizable on a real robot. The authors argue that we can use a segmentation algorithm to track the object being thrown and use an impedance controller to implement the movement on an arm with a Shawdow Hand as its end-effector. However  designing such a controller is a very difficult problem  because the impedance must depend on the object s physical properties and the hand  as well as the release velocity of the object being thrown. It is unclear how these quantities can be obtained in the real world. Furthermore  tracking an object in the air using perception foundation models (e.g. FoundationPose) seems unreasonable -- it would only work in a setup where the object is moving at a speed where such models can make inferences at a matching frequency. These models however are generally large and slow.   On the bright side  the paper tackles an important problem of dynamic catching of objects and has an interesting approach. Further experiments addressing concerns around real-world implementation in the camera ready version will strengthen the paper.

===========================================  Reviewers appreciate that the paper tackles the problem of dynamic catching  which is an important yet understudied problem in robotics. They found the proposed method clever and interesting. However  reviewers are concerned whether this can be realized in the real world -- especially with the unrealistic motions it generates. A real-world experiment would resolve these concerns.